# BTK Isoforms p80 and p65 Are Expressed in Head and Neck Squamous Cell Carcinoma (HNSCC) and Involved in Tumor Progression

**DOI:** 10.3390/cancers15010310

**Published:** 2023-01-03

**Authors:** Annika C. Betzler, Hannah Strobel, Tsima Abou Kors, Jasmin Ezić, Kristina Lesakova, Ronja Pscheid, Ninel Azoitei, Johanna Sporleder, Anna-Rebekka Staufenberg, Robert Drees, Stephanie E. Weissinger, Jens Greve, Johannes Doescher, Marie-Nicole Theodoraki, Patrick J. Schuler, Simon Laban, Toshiro Kibe, Michiko Kishida, Shosei Kishida, Christian Idel, Thomas K. Hoffmann, Marialuisa Lavitrano, Emanuela Grassilli, Cornelia Brunner

**Affiliations:** 1Department of Oto-Rhino-Laryngology, Ulm University Medical Center, 89075 Ulm, Germany; 2Department of Internal Medicine I, Ulm University Medical Center, 89081 Ulm, Germany; 3Institute of Pathology, Ulm University Medical Center, 89081 Ulm, Germany; 4Department of Biochemistry and Genetics, Graduate School of Medical and Dental Sciences, Kagoshima University, Kagoshima 890-8580, Japan; 5Department of Otorhinolaryngology, University Hospital Schleswig-Holstein, University of Luebeck, Campus Luebeck, 23538 Luebeck, Germany; 6School of Medicine and Surgery, University of Milano-Bicocca, 20900 Monza, Italy

**Keywords:** BTK, head and neck cancer, AVL-292, ibrutinib

## Abstract

**Simple Summary:**

Bruton’s Tyrosine Kinase (BTK) was originally considered to be primarily expressed in cells of hematopoietic origin. Apart from the 77 kDa BTK isoform expressed in immune cells, elevated expression of novel BTK isoforms of 80 and 65 kDa have been recently described for several solid tumor entities. These newly described isoforms have been linked to tumor growth and poor prognosis. Therefore, we aimed to investigate whether BTK isoforms are also expressed in head and neck squamous cell carcinoma (HNSCC) and further the molecular and cellular consequences of BTK expression for HNSCC tumorigenesis. We confirmed the expression of the BTK-p65 and BTK-p80 isoforms in HNSCC and revealed that both isoforms are products of the same mRNA. Abrogation of BTK activity inhibited tumor progression in our study. Thus, targeting BTK activity appears as a promising therapeutic option for patients suffering from BTK expressing HNSCC.

**Abstract:**

Here, we describe the expression of Bruton’s Tyrosine Kinase (BTK) in head and neck squamous cell carcinoma (HNSCC) cell lines as well as in primary HNSCC samples. BTK is a kinase initially thought to be expressed exclusively in cells of hematopoietic origin. Apart from the 77 kDa BTK isoform expressed in immune cells, particularly in B cells, we identified the 80 kDa and 65 kDa BTK isoforms in HNSCC, recently described as oncogenic. Importantly, we revealed that both isoforms are products of the same mRNA. By investigating the mechanism regulating oncogenic BTK-p80/p65 expression in HNSSC versus healthy or benign tissues, our data suggests that the epigenetic process of methylation might be responsible for the initiation of BTK-p80/p65 expression in HNSCC. Our findings demonstrate that chemical or genetic abrogation of BTK activity leads to inhibition of tumor progression in terms of proliferation and vascularization in vitro and in vivo. These observations were associated with cell cycle arrest and increased apoptosis and autophagy. Together, these data indicate BTK-p80 and BTK-p65 as novel HNSCC-associated oncogenes. Owing to the fact that abundant BTK expression is a characteristic feature of primary and metastatic HNSCC, targeting BTK activity appears as a promising therapeutic option for HNSCC patients.

## 1. Introduction

Cancer is characterized by a large array of genetic alterations causing cellular and molecular abnormalities that allow cells to escape their controlled life cycle. In human cancers, the cell transformation process is often associated with dysregulation of tyrosine kinases, which are key players in most signal-transduction pathways governing cellular proliferation, survival, differentiation, and motility. Most dysregulation of tyrosine kinases occurs through inappropriate expression, activation, or both [1]. One tyrosine kinase often found to be dysregulated in cancer is Bruton’s tyrosine kinase (BTK).

BTK is a non-receptor tyrosine kinase belonging to the TEC family. Originally, BTK was discovered as the product of the defective gene in X-linked agammaglobulinemia (XLA) almost 30 years ago [2,3]. BTK is involved in a broad range of immunological pathways, playing a central role in innate and adaptive immunity [4]. Shortly after its discovery, the essential role of BTK for B cell development and function became evident. BTK is a key component of the BCR signaling pathway [5]. Moreover, it is involved in further pathways including chemokine receptor, Toll-like receptor (TLR) or Fc receptor signaling [6]. Besides B cells, BTK is also expressed in cells of myeloid origin including macrophages, monocytes, neutrophils, dendritic cells (DCs) and mast cells [4]. In many B cell malignancies, like mantle cell lymphoma, chronic lymphocytic leukemia, or Waldenström macroglobulinemia, dysregulation of B cell receptor signaling occurs upon constitutive activation of BTK [7]. Therefore, BTK inhibitors such as FDA-approved Ibrutinib emerged as an option in preventing survival of malignant B cells.

Apart of the hematopoietic system, elevated expression of BTK was recently reported in several solid tumor entities, like breast [8], ovarian [9,10], prostate [11,12], colon [13,14], gastric carcinoma [15], glioma [16,17,18] as well as non-small-cell lung carcinoma (NSCLC) [19].

Interestingly, whereas in B cells a 77 kDa isoform is expressed (also called BTK-A because of the expression of exon 1A; [8]), novel BTK isoforms of 80 and 65 kDa are expressed in solid tumors. In particular, the 80 kDa isoform has been reported in breast and prostate carcinomas [8,11], whereas BTK-p65 has been shown to be abundantly expressed in colon [13,14] and ovarian carcinomas [10], glioblastomas [18] and NSCLC [19]. The BTK-p80 isoform (also called BTK-C because of the expression of exon 1C; [8]) is expressed from an alternative promoter and the mRNA contains an alternative first exon (exon 1C). This is located upstream of exon 1 from the BTK-p77 transcript found in B cells (Figure 1A; [8]) and contains two start codons. The use of the first start codon leads to an extension of 34 amino acids at the amino-terminal end, which results in the expression of a protein with a molecular weight of 80 kDa [8]. The BTK-p80 isoform is spliced from a donor site at the end of exon 1C and an acceptor site at the beginning of exon 2. Therefore, the sequence of exon 1A of the BTK-p77 isoform is not present in the BTK-p80 protein [8]. According to the published literature, the shorter BTK-p65 isoform is also expressed from the same alternative exon as BTK-p80 [13]. Exon 1b is located upstream from the first exon of BTK-p77 (Figure 1A). However, the translation of Btk-p65 starts from a start codon in exon 4, which leads to a protein with a molecular weight of 65 kDa (Figure 1A) [13]. This isoform lacks most of the amino-terminal pleckstrin homology (PH)-domain, important for recruitment to the cell membrane, a prerequisite for its phosphorylation and therefore BTK activation in hematopoietic cells (Figure 1B) [20]. Thus, a different regulation of BTK-p65 activity in tumor cells is expected. The translation of BTK-p65 is mediated by the interaction with homo sapiens heterogeneous nuclear ribonucleoprotein K (hnRNPK) [13]. Due to the binding of hnRNPK to the 5′ untranslated region (UTR), the mRNA adopts a complex secondary structure in which the first start codon is hidden in a hairpin loop. This probably favors translation from the start codon in exon 4. The RNA-binding nuclear protein hnRNPK can shuttle between the nucleus and cytoplasm. The subcellular localization of hnRNPK is regulated by ERK1/2, thus influencing BTK-p65 translation. Furthermore, the translation depends on an internal ribosomal entry site (IRES) located in the 5′UTR of the mRNA [13].

In solid tumor models, the expression of BTK is associated with tumor growth, protection from apoptosis and poor prognosis of cancer patients. RNA interference- or pharmacologic-mediated abrogation of BTK expression/activity negatively impacted on cell proliferation and induced apoptosis [8,9,11,12,13,15,16,17]. Therefore, BTK was suggested as a potential therapeutic target for treatment of solid BTK-expressing tumors.

The present study aims to investigate the expression of BTK isoforms in head and neck squamous cell carcinoma (HNSCC). HNSCC is the sixth most common tumor disease worldwide [21]. HNSCCs develop from the mucosal epithelium in the oral cavity, pharynx and larynx and are the most common malignancies that arise in the head and neck [22]. Main risk factors for the development of HNSCC include alcohol, smoking, and infection with the human papillomavirus (HPV). Tumors arising in the oropharynx are increasingly linked to HPV infection, while tumors of the oral cavity and larynx are still primarily associated with smoking and alcohol [22]. In general, HPV-positive tumors are associated with a better prognosis compared to HPV-negative tumors. Current treatment options mainly include surgery, radio- and/or chemotherapy. Although new treatment options (immune checkpoint blockade, EGFR inhibition) have been added in the past years, mortality and recurrence rates remain high [23]. Therefore, the need for new targeted therapy approaches is high. Consequently, we aimed to investigate whether treatment of HNSCC with BTK inhibitory drugs would be a beneficial new therapeutic option.

Having observed BTK-p80 as well as BTK-p65 expression in HNSCC cell lines and tumor specimens we further investigated the molecular and cellular consequences of this finding for HNSCC tumorigenesis. Together, our in vitro and in vivo data suggest the abrogation of BTK expression or its activity in patients with HNSCC as a powerful treatment option to suppress tumor growth as well as tumor induced angiogenesis.

## 2. Materials and Methods

### 2.1. Cell Lines and Cell Culture

The following HNSCC cell lines were used (the study was approved by the local Ethics Committee #219/17) including UD (University of Düsseldorf; source: Henning Bier) -SCC-1, SCC-2, SCC-3, SCC-5, SCC-6, SCC-8 and UM (University of Michigan; source: Thomas E. Carey) -SCC-10A, SCC-11B, SCC-17B and SCC-22B [24,25]. HNSCC cell lines were cultured as described [26]. The Burkitt lymphoma cell lines (source: ATCC) Namalwa and Ramos as well as the B cell lymphoma cell line MHH-PREB1 (DSMZ no.: ACC-354) and the prostate cancer cell lines (source: ATCC) DU145, PC-3 and LNCaP were cultured using RPMI (Life Technologies GmbH) supplemented with 10% FBS and 1% penicillin/streptomycin at 37 °C and 5% CO_2_. The non-tumor oral epithelial cell lines MOE1a and MOE1b were cultured as described [27]. The hTERT TIGKs cell line was obtained from ATCC (CRL3397) and handled as suggested [28]. The identity of all cell lines has been regularly verified by STR analyses over the last three years. Sequencing of TP53 gene mutations additionally validated the identity of HNSCC-derived cell lines [24,25]. The HST116p53KO cells were a kind gift of Dr. B. Vogelstein (Johns Hopkins University, Baltimore, MD, USA). All experiments were performed with mycoplasma-free cells. Further details of the cell lines are shown in Appendix A.

### 2.2. RNA Isolation

RNA was prepared using the RNeasy Mini Kit and the QIAshredder spin columns from Qiagen (Qiagen, Venlo, The Netherlands) according to the manufacturer’s protocol.

### 2.3. Quantitative Expression Analysis of BTK-p77 and BTK-p80/p65

Relative mRNA expression was quantified by QuantiNova SYBR Green RT-PCR assay (Qiagen, Venlo, The Netherlands). The following primers were used: BTK-p77 fw 5′-TCC TTC CTC TCT GGA CTG TAA GAA TAT-3′, rev ACT TGG AAG GTG GGA CTC GAT; BTK-p80/p65 fw 5′-TCT GCT ACG TAG TGG CGT TC-3′, rev 5′-GGA GGA CCC CTT CAT TCG AC-3′; RPL13 fw 5′-CGG ACC GTG CGA GGT AT-3′, rev 5′-CAC CAT CCG CTT TTT CTT GTC-3′; HPRT fw 5′-GAC CAG TCA ACA GGG GAC AT-3′, rev 5′-GTG TCA ATT ATA TCT TCC ACA ATC AAG-3′, which were obtained from biomers.net GmbH. Gene-specific PCR products were measured with the LightCycler 96 Instrument (Hoffmann- La Roche AG). For relative quantification the 2−ΔΔCT-method was used [29].

### 2.4. Isolation of CD19^+^ B Cells from PBMC

To validate specificity of BTK antibodies CD19^+^ B cells were isolated from PBMCs of healthy donors. PBMCs were isolated from blood which was anticoagulated with citrat by using Leukosep-Tubes (Greiner bio one, Kremsmünster, Austria) and the separating medium Biocoll (Merck/Biochrom, Berlin, Germany). CD19^+^ B cells were isolated from human PBMCs by negative selection using the EasySep™ Human B Cell Isolation Kit (Stemcell, Vancouver, Canada) according to manufacturer’s instructions.

### 2.5. Immunoblot Analysis

Immunoblots were performed according to standard procedures using total protein extracts. Membranes were stained with anti-BTK-p80 7F12H4 (Santa Cruz), BTK-p80 DFS (Santa Cruz), BTK-p65 [13], cyclin-D1 (CCND1), pRB1 (Ser780), Bcl-xl (BCL2L1), caspase 9 (CASP9), LC3A/B-I (MAP1LC3A/B), LCA/B-II and ATG7 antibodies (all obtained from Cell Signaling) and anti-β-actin (AKTB) monoclonal mouse antibody (Sigma-Aldrich, Buenos Aires, Argentina) and GAPDH (Santa Cruz, Dallas, TX, USA). Horseradish-peroxidase conjugated secondary antibodies were purchased from Thermo-Scientific. Membranes were developed with a chemiluminescence detection system (Bio-Rad, Hercules, CA, USA).

#### Research Subjects

Corresponding primary cancer tissue specimens of the head and neck region and healthy non-tumor squamous epithelial cells were obtained from two treatment-naive HNSCC patients with histologically confirmed tumors of the oral cavity who were treated at the Department of Otorhinolaryngology, Head and Neck Surgery, Ulm University in 2016 and 2018. Tissue collection was approved by the Ethics Committee of the University of Ulm (#90/15) and each patient provided informed consent. Further clinical data is provided in Appendix A.

### 2.6. Proliferation Assay

Cell proliferation was analyzed using the CellTiter 96 AQueous One solution Cell Proliferation Assay (Promega Corporation, Fitchberg, WI, USA). Cells were seeded on 96-well plates and grown at 37 °C and 5% CO_2_ in complete DMEM (10% FBS). After 24 h, fresh medium was given (10% FBS) containing the indicated inhibitors or DMSO for the time indicated. The absorbance was measured with a Tecan Infinite M200 Pro microplate reader (Tecan Group AG, Zürich, Switzerland). The experiments were performed once in triplicates.

### 2.7. Cell Cycle Analysis

Cells were harvested using trypsin/EDTA solution and washed with PBS. For cell cycle analyses cells were fixed overnight using ice-cold 70% ethanol, spun down and incubated for 1 h at 37 °C in the presence of 50 µg/mL propidium iodide and 100 µg/mL RNAse A in PBS. The cell cycle distribution was recorded by Flow Cytometry (FACS Calibur, Becton Dickinson, Franklin Lakes, NJ, USA) and analyzed using ModFit LT software (Verify Software House, Topsham, MA, USA).

### 2.8. Chorio-Allantoic Membrane (CAM) Assay and IHC Analysis

Chorio-allantoic membrane (CAM) assays were performed as described [30]. Briefly, 1.5 × 10^6^ cancer cells were applied within 5 mm silicon rings on the surface of the chicken CAM, 8 days after egg fertilization. Xenografts were either treated with DMSO or with the BTK inhibitor AVL-292 (10, 20 or 30 μM) 24 and 48 h after tumor xenograft. Four days after implantation tumors were harvested and subjected for analyses as described previously [26].

### 2.9. Short Hairpin RNA Vector Cloning and Lentiviral Infection

The sequence of BTK shRNA (5′-ACCGG-GCGGAAGGGTGATGAATATTT-GTTAATATTCATAGC-AAATATTCATCACCCTTCCGC-TTTTTTg-3; 5′-aattCAAAAAA-GCGGAAGGGTGATGAATATTT-GCTATGAATATTAAC-AAATATTCATCACCCTTCCGC-C-3′) was obtained from the Broad Institute GPP Web Portal TRC2 library (http://portals.broadinstitute.org/gpp/public/clone/details?cloneId=TRCN0000065334 accessed on 9 May 2017) and was cloned into the BbsI and EcoRI sites of the pRSI12-U6-sh-UbiC-TagRFP-2A-puro vector (Cellecta [https://www.cellecta.com/] accessed on 9 May 2017, obtained from BioCat [Heidelberg, Germany]). The scrambled shRNA, which does not target any human genomic sequence, was described elsewhere [26] and served as internal control. BTK shRNA as well as scrambled control shRNA were overexpressed using the lentiviral pRSI12-U6-sh-UbiC-TagRFP-2A-puro vector as described previously [26].

### 2.10. Scratch Assay/Wound Closure Assay

A total of 1 × 10^5^ cells per cm^2^ were seeded into culture-insert 2 Well (Ibidi) placed on 12-well plates in DMEM supplemented with 10% FBS and cultivated at 37 °C and 5% CO_2_ for 24 h. After appropriate cell attachment, culture inserts were removed creating a cell-free gap of approximately 500 µm. Photomicrograph images (Axio Observer D1 fluorescence microscopy, Carl Zeiss AG) were immediately captured (time 0 h), and the cells were subsequently incubated in DMEM (1% FBS). The migration of the cells and the closing of the scratch were observed by taking microphotographs. The wound area was measured by using ImageJ (NIH, Bethesda, MD, USA) software. Three experiments were performed in triplicate.

### 2.11. Transwell Migration Assay

The transmigration capacity of HNSCC-derived cell lines was assessed by the xCELLigence real time cell analysis system (OMNI Life Science, East Taunton, MA, USA). Electrodes of transwells were coated with 30 µL of a 50 µg/mL fibronectin solution for 30 min at RT. 20.000 cells/well were seeded in complete DMEM (10% FBS). The BTK inhibitor AVL-292 was added in different concentrations as indicated. DMSO treatment served as internal control. Transmigration was monitored over time. Three independent experiments for each cell line were performed in quadruplicates.

### 2.12. VEGFA ELISA

VEGFA ELISA was performed as described previously [26].

### 2.13. TCGA Dataset & Data Analysis

The TCGA BTK and RLP36A DNA Methylation dataset of HNSCC (n = 528) and paired healthy mucosa (n = 50) were downloaded from http://maplab.imppc.org/wanderer in November 2021 accessed on 9 November 2021. The values used were Beta values obtained from the 450 k Methylation array. Data analysis was performed in R (4.1.1). Kruskal-Wallis test was performed for comparing Beta values of the BTK alternative promoter CpG loci between the two groups using the stats (4.1.1) package. Dunn test with BH correction was then performed as a Post Hoc using rstatix (0.7.0) package. Visualization was done using the ggplot2 (3.3.5) package. The map was generated using the chromoMap (0.3.1) package.

### 2.14. Statistical Analysis

Results are illustrated as arithmetic means ± standard deviation. Data were tested for normality and accordingly analyzed either with a two-tailed unpaired Student’s *t*-test or Mann-Whitney-U test as indicated, with GraphPad Prism V8 software. * *p* < 0.05, ** *p* < 0.01, *** *p* < 0.001, **** *p* < 0.0001.

## 3. Results

### 3.1. BTK-p80 and BTK-p65 Isoforms Are Expressed in HNSCC-Derived Cell Lines

Recently, two novel BTK isoforms of 65 and 80 kDa were identified in solid tumors [8,11,13]. To investigate, whether these isoforms are also present in tumors of the head and neck origin, we developed a RT-PCR design allowing the discrimination between BTK-p77, BTK-p80 and BTK-p65 mRNA transcripts. Examination of nucleotide sequences revealed substantial similarity between the BTK-p80 and BTK-p65 isoforms. The sequence alignment using ClustalX 2.1 (Figure 1C) confirmed that BTK-p80 (accession: NM_001287344, mentioned in [8]) and BTK-p65 sequences [13] are identical. Both isoforms contain the same first exon which is different from the first exon of BTK-p77 (accession: NM_000061.2 or AH006678 (only the exons); old: U113399, mentioned in [8]). The sequences coding different BTK isoforms are identical toward 3′ end starting with exon 2. In conclusion, BTK-p80 and BTK-p65 are coded by the same transcript and cannot be distinguished on mRNA level (Figure 1C). ‘Isoform’ specific primers targeted to each first exon allow just the differentiation of the BTK-p77 and BTK-p80/p65 mRNA (Figure 1C). Thus, isoform specific primers were used for quantitative RT-PCR of ten human HNSCC-derived cell lines and three prostate carcinoma cell lines, the latter serving as internal controls for oncogenic BTK expression [11]. The Burkitt lymphoma cell line Namalwa served as an internal control for BTK-p77 expression. The samples were normalized to the house keeping genes RPL13 and HPRT, whereas the malignant B cell line Namalwa was set as calibrator. BTK-p80/p65 mRNA was identified in Namalwa B cells in which only the expression of BTK-p77 was expected. In contrast, in cancer cells derived from solid tumors used in this analysis (HNSCC and prostate carcinoma), only BTK-p80/p65 mRNA was detectable, albeit at different expression levels (Figure 1D,E). In order to delineate whether either BTK-p80 or BTK-p65 isoform or both are expressed in HNSCC cell lines, protein expression was analyzed. For this, several commercially available BTK-antibodies were tested for their ability to detect specific isoforms. We observed that the BTK antibody 7F12H4 (Santa Cruz Biotechnology, Dallas, TX, USA, (sc-81159)) detects only the BTK-p80 isoform, whereas the BTK antibody DFS (Santa Cruz Biotechnology; sc-81735) detects both, the BTK-p77 as well as the BTK-p80 isoform (Figure 1F). Healthy B cells isolated from human PBMCs were used as control for the expression of BTK-p77, which was only detected with the BTK DFS antibody (Figure 1F). The BTK-antibody E-9 (Santa Cruz Biotechnology; sc-28387) detects BTK-p77 only (data not shown). None of the commercially available and tested antibodies detect the BTK-p65 isoform, but the previously isotype-specific BN49 BTK antibody [13]. By using these antibodies considerable BTK-p80 and BTK-p65 expression levels in all tested HNSCC cell lines were detected (Figure 1G,H). In contrast, the BTK-p80 isoform was not detectable in Moe1a, Moe1b and TIGK cells, which represent non-tumor cell lines derived from oral epithelium (Appendix A). Additionally, BTK-p80 expression was detected in primary cancer tissue specimens of the head and neck region, but not in non-tumor squamous epithelial cells obtained from the same individuals (Appendix A).

### 3.2. BTK Inhibition Affects Proliferation, Transmigration and VEGFA Secretion of HNSCC Derived Cell Lines

To explore potential effects of BTK inhibition in HNSCC cells, 10 different HNSCC cell lines were subjected to the two BTK inhibitors Ibrutinib and AVL-292. Additionally, the PI3K inhibitor LY294002 was used in preventing BTK phosphorylation and thereby its activation [31,32]. To investigate the effect of BTK inhibition on proliferation, HNSCC cells were treated with four different concentrations of respective inhibitors based on published data [11,13] and cell number was assessed after 24 h, 48 h and 72 h. Treatment with all three BTK inhibitors negatively impacted the proliferation in a time- and dose-dependent manner for all analyzed HNSCC cell lines, particularly after 72 h (Appendix A). Since treatment with AVL-292 resulted in the strongest decrease in proliferation, this inhibitor was used for all subsequent experiments. Having demonstrated that BTK activity is essential for HNSCC proliferation, we next analyzed the role of the kinase in transmigration. Cell lines were selected based on their BTK-p80 expression and their responsiveness to BTK inhibitors revealed by the proliferation assay. We aimed to include cell lines that cover the whole spectrum of Btk expression and responsiveness to BTK inhibitors. Consequently, we used UDSCC1 and UDSCC2 (intermediate BTK expression, intermediate responder), UDSCC5 (high BTK expression, strong responder) and UDSCC6 (lower BTK expression, low responder) for subsequent experiments. The xCELLigence real time cell analysis system revealed a considerable and significant decrease in the transmigratory capacity of UDSCC1, UDSCC5 and UDSCC6 cell lines after 24 h, 48 h and 72 h of BTK inhibition with AVL-292 (Figure 2A). Fast proliferation of cancer cells is associated with a high demand of oxygen and nutrients. This demand is secured by an intense tumor-driven vascularization in which the vascular endothelial growth factor A (VEGF A) plays a crucial role. Therefore, we tested whether BTK activity is also necessary for VEGFA secretion by HNSCC cells. BTK inhibition in UDSCC5 cells, which secrete the highest amounts of VEGFA in comparison to all other tested HNSCC-derived cell lines (data not shown), resulted in a distinct reduction in VEGFA secretion after 96h of treatment with AVL-292 (Figure 2B). These data indicate that BTK inhibition by AVL-292 impairs proliferation, migration and VEGFA secretion in HNSCC-derived cell lines.

### 3.3. BTK Inhibition Induces Cell Cycle Arrest, Apoptosis and Autophagy in HNSCC-Derived Cell Lines

To get further insight into the mechanism by which BTK inhibition reduces HNSCC cell proliferation, we additionally investigated cell cycle, apoptosis and autophagy of selected HNSCC cell lines after BTK inhibition with AVL-292.

Cell cycle analyses of UDSCC5 cells revealed a G2/M arrest at the expense of G0/G1 and S phase upon BTK inhibition with AVL-292 both 24 h and 48 h post treatment (Appendix A). In UDSCC6 cells, BTK inhibition resulted in G2/M arrest at 24 h and Go/G1 at 48 h after treatment with AVL-292. (Appendix A). The pronounced cell cycle arrest was associated with significantly reduced cyclin D1 (CCND1) expression, impaired phosphorylation of retinoblastoma protein (RB1) and reduced glycogen synthase kinase 3 beta (GSK3B) expression on protein level in UDSCC1, UDSCC2 and UDSCC6 cell lines upon BTK inhibition (Figure 3A,B).

Additionally, we observed an increase in the number of apoptotic cells in both UDSCC5 and UDSCC6 cell lines 48 h after BTK inhibition (Appendix A–E). This observation was accompanied with enhanced levels of the pro-apoptotic protein cleaved caspase-9 (cleaved CASP9) indicating that cell death occurs via the mitochondrial pathway. Furthermore, BTK inhibition was corroborated with down-regulation of anti-apoptotic protein Bcl-xl (BCL2L) (Figure 3C,D).

Inhibition of BTK by AVL-292 also increased autophagy events as demonstrated by augmented expression of the autophagy markers ATG7, LC3A/B-I and LC3A/B-II in HNSCC cell lines (Figure 3E,F).

Taken together, these data show that BTK inhibition by AVL-292 negatively impacts HNSCC proliferation by inducing cell cycle arrest, apoptosis and autophagy in tumor cells.

### 3.4. BTK Inhibition Impairs Tumor Growth and Angiogenesis In Vivo

Since we revealed that BTK inhibition reduces HNSCC autonomous cell growth in vitro, we next aimed to examine its effect in tumor progression in vivo. Therefore, we used the chicken chorio-allantoic membrane (CAM) assay. UDSCC6 cell line was chosen for CAM assay as this cell line revealed a rather low BTK expression (Figure 1G) and low responsiveness to BTK inhibition in the proliferation assay (Appendix A).

UDSCC6 cells were applied to the CAM on day 8 after fertilization and allowed to form tumors for the next 94 h. AVL-292 was applied ectopically onto the tumors 24 and 72 h after tumor xenograft. BTK inhibition resulted in a dose-dependent reduction in tumor formation (Figure 4A,B). In addition, treatment with AVL-292 significantly reduced the size of developing tumors and was corroborated with decreased number of proliferating Ki67-positive tumor cells (Figure 4C,D,F). Importantly, tumor xenografts revealed a significant reduction in desmin-positive pericytes upon BTK inhibition, indicating a strong reduction in tumor-induced vascularization (Figure 4D,E). The reduced vascularization documented on in vivo tumor xenografts is in line with the in vitro impaired VEGF A secretion upon BTK inhibition (Figure 2B). Taken together, these data suggest BTK’s double-pronged role in cancer cell proliferation on the one side and tumor-driven angiogenesis on the other, both events acting ultimately toward a sustained tumor growth. Additionally, our data demonstrate that even cells characterized by a lower response to BTK inhibition reveal impaired tumor growth and angiogenesis in vivo induced by BTK activity abrogation.

### 3.5. BTK Inhibition and Its Genetic Abrogation Delayed Wound Closure of HNSCC-Derived Cell Lines

In order to validate our inhibitor data described so far, BTK knock-down experiments were conducted. UDSCC1 cells were transduced with lentiviruses, expressing either scrambled shRNA (scr-shRNA) or shRNA targeting BTK (BTK-shRNA). We confirmed abrogation of BTK expression in UDSCC1 cells by immunoblot analysis (Figure 5A). Knock-down of BTK clearly delayed wound closure in scratch assays performed with UDSCC1 cells (Figure 5B,C). These findings complement the effect of BTK on the migratory activity and confirm the oncogenic function of BTK in HNSCC.

### 3.6. Association between BTK Expression and Methylation

Since we demonstrated an abundant expression of the oncogenic BTK-p80/p65 isoform in HNSCC cell lines as well as in primary HNSCC samples, we next investigated the mechanisms controlling BTK-p80/p65 expression in HNSCC cells. As epigenetic alterations like changes in DNA methylation are common characteristics of cancer cells, we wondered whether BTK-p80/p65 expression could possibly be regulated by methylation in HNSCC.

We identified six CpG loci in the alternative promoter (described in [8]) upstream of Exon 1C (Figure 6A), where the transcription of the messenger encoding for BTK-80/p65 isoforms starts. We analyzed the methylation level of the desired loci in the TCGA HNSCC patients (n = 528) and paired healthy mucosa (n = 50). When comparing beta-values, we detected a significant reduction in the methylation level in tumors compared to paired healthy mucosa for all six CpG loci examined (Figure 6B). Loci cg00126698, cg02998933, and cg21909660 are present in the 5′ UTR and cg03791917, cg08355317, and cg16397002 in the transcriptional start site (TSS) of the alternative BTK-p80 promoter. Our epigenetic findings suggest that in non-cancerogenic tissues the BTK-p80/p65 promoter is characterized by a higher degree of methylation and therefore likely less active. In contrast, in tumor cells this region is less methylated and therefore could be more active, resulting in an upregulation of BTK-p80 and BTK-p65 protein expression under cancerogenic conditions.

## 4. Discussion

The heterogeneity of HNSCCs impedes both the identification of specific targets as well as the development of targeted therapy approaches. There is an urgent need for more effective therapies alongside clinically relevant biomarkers to improve patients’ outcome. Understanding the molecular mechanisms of HNSCC represents the basis for identification of specific molecules, allowing early detection, targeted therapy, monitoring and prediction of prognosis. Recently, expression of BTK-p65 and BTK-p80 isoforms was described in several solid tumor entities [8,9,10,11,12,13,14,15,16,17,18,19]. In search of novel molecular markers adding new treatment options of HNSCC we investigated whether oncogenic BTK is also expressed in the head and neck region tumors.

Recently, an aberrantly high expression of BTK was observed in specimens of oral squamous cell carcinoma (OSCC) patients together with reduced migration and invasion upon Ibrutinib treatment [33]. However, which BTK isoform is involved in OSCC has not been investigated in detail. Since the authors used the 7F12H4 antibody for detection of BTK as we did in our study it is very likely that the BTK-p80 isoform was revealed in analyzed OSCC samples. We show here, that both BTK-p65 as well as BTK-p80 isoforms, but not the hematopoietic BTK-p77 isoform, are expressed in various HNSCC cell lines as well as in primary HNSCC tissue. No co-expression of both oncogenic BTK isoforms in one tumor entity has been described so far. Interestingly, our in-silico analysis revealed that both, the BTK-p80 and BTK-p65 isoforms, are produced by the same mRNA. The employment of isoform-specific antibodies revealed considerable expression of both oncogenic isoforms on protein level. Isoform-specificity of antibodies was proven by using appropriate controls including healthy B cells, multiple B cell lymphoma cell lines, non-tumor cell lines derived from oral epithelium as well as healthy non-tumor squamous epithelial cells.

Our data characterize BTK-p65 and BTK-p80 as novel HNSCC-associated oncogenes promoting HNSCC cell survival. Since AVL-292 abolishes kinase activity of both isoforms, the precise contribution of each isoform to oncogenic processes in HNSCC could not be distinguished so far. However, BTK-specific inhibition decreased HNSCC proliferation by inducing cell cycle arrest, apoptosis, and autophagy, making the kinase a promising therapeutic target for patients with BTK expressing HNSCC. In addition, we substantiated our in vitro data with investigations conducted in an in vivo animal experimental model. Here, we could demonstrate that BTK inhibition by AVL-292 impaired tumor growth and its vascularization in tumor xenografts. Altogether, our findings suggest a dual role of BTK during tumorigenesis and tumor progression. Our findings are in line with several studies reporting similar results in prostate, breast, and colon cancer [8,11,13]. AVL-292, currently in phase 1 clinical trials for hematological malignancies, binds to BTK with high specificity and inhibits its activity, which subsequently impacts on PLCγ2 phosphorylation [34,35]. So far, the exact mechanisms for BTK-p80 and BTK-p65 –dependent tumor cell growth are unknown and might involve pathways different from those reported in hematopoietic cells. BTK-p65 isoform lacks most of the amino terminal PH-domain. Therefore, it is likely that the activity of these oncogenic isoforms is differentially regulated and activated than the hematopoietic BTK-p77 isoform [8,13]. The PH-domain allows BTK translocation to the plasma membrane, necessary for its phosphorylation and activation [20]. The PH-domain is also required for interaction with other proteins, especially negative regulators of BTK activity [36,37,38]. As a result, lacking the region responsible for negative regulation, could explain aberrant expression and activation of BTK-p65 in cancer. Interestingly, it has been reported that loss of part of PH-domain leads to a decrease in the auto-inhibited form of BTK and higher levels of spontaneous auto-activation [39].

In B cells, the BTK-p77 isoform is activated upon B cell receptor engagement followed by downstream phosphorylation of LYN and SYK. Subsequently, important signal transduction molecules like PLCγ, AKT, κAT and NF-κB regulate proliferation, cell cycle or apoptosis [40,41]. Our here described findings suggest that the oncogenic BTK-p65 and BTK-p80 isoforms also provide signals that regulate proliferation, cell cycle and apoptosis in HNSCC, although the exact mechanisms are incompletely understood. However, in TPA-induced breast cancer cells BTK is also involved in the PLCγ-dependent NF-κB activation, leading to the upregulation of metalloproteinase-9, crucially involved in cancer cell invasion [42]. Therefore, it seems promising to study the PLCγ-NF-κB pathway in HNSCC to understand the molecular consequences of BTK expression and its oncogenic potential. Additionally, BTK-p65 level and its transforming potential seem to depend on the RAS/ERK/MAPK pathway in colon cancer [13] and also in NSCLC [19]. One of the most frequent alterations found in HNSCC is the aberrant expression of the epidermal growth factor receptor (EGFR), upstream of the RAS/ERK/MAPK signaling pathway [43,44], which is consequently, deregulated in HNSCC [45,46,47]. Thus, a similar association of BTK-p65 expression and RAS/ERK/MAPK signaling in HNSCC could be envisaged.

Eiffert et al. [8] described that the BTK-p80 isoform is expressed from an alternative promoter and transcribed from an alternative first exon upstream from exon 1 found in the BTK-p77 transcript. Given that both oncogenic BTK isoforms p80 and p65 derive from the same mRNA, the transcriptional regulation of BTK-p80 and BTK-p65 expression is expected to be similar. The expression of BTK-p65 is further regulated at translational level [13]. The aberrant use of alternative promoters has been directly linked to cancer cell growth. Thus, previous reports revealed increased alternative promoter usage in tumor cells compared to healthy cells [48,49,50,51]. Deregulation of its alternative promoter might cause increased expression of the BTK-p80/p65 isoforms in HNSCC observed in our study. Our epigenetic analysis of the TCGA cohort suggests that the alternative promoter of BTK-p80/p65 could be regulated by methylation. Our analysis reveals a relatively hypomethylated profile of the CpG loci within the alternative BTK-p80 promoter in HNSCC tumors compared to healthy mucosa. The higher methylation level identified in healthy mucosa cells at these sites seems to be consistent with the transcriptional suppression of BTK-p80. While in HNSCC patients, these CpG sites seem to become demethylated, resulting in BTK-p80/p65 protein expression. However, since the whole tumor mass (cancer cells plus immune cell infiltrate) was used for TCGA cohort RNA-seq, correlation of methylation with expression level of the oncogenic BTK isoforms was impossible, as we cannot distinguish the different BTK-isoforms (BTK-p77 expressed by hematopoietic cells versus BTK-p65/p80 expressed in tumor cells) based on RNA-seq data. Analyzing BTK expression in the TCGA OSCC cohort, Liu et al. [33] reported an improved overall survival of patients with high BTK expression. The interpretation of these data is rather intricate, since BTK is highly expressed in tumor infiltrating B and innate immune cells. Therefore, it is difficult to distinguish whether the signal derived from BTK in tumor or from BTK in immune cells. A parallel investigation of tumor tissue sections using the appropriate isoform-specific antibodies would have been helpful. However, the authors revealed a high BTK expression in TCGA OSCC patients with inoperable or locally advanced unresectable disease receiving concurrent chemoradiotherapy (CCRT), which was associated with a significantly worse prognosis. It would be interesting to know which BTK isoform is expressed in these types of oral cancer.

## 5. Conclusions

Altogether, our findings show the expression of oncogenic BTK-p65 and BTK-p80 isoforms in HNSCC. Moreover, abrogation of BTK activity inhibited tumor progression in our study in terms of proliferation and vascularization in vitro and in vivo. These observations were associated with cell cycle arrest and increased apoptosis and autophagy. Altogether, the here presented data suggest that targeting BTK activity could be a promising therapeutic option for HNSCC patients, either as monotherapy or possibly in combination with current treatment options like target therapy, radiotherapy or immune checkpoint inhibition.

## Figures and Tables

**Figure 1 cancers-15-00310-f001:**
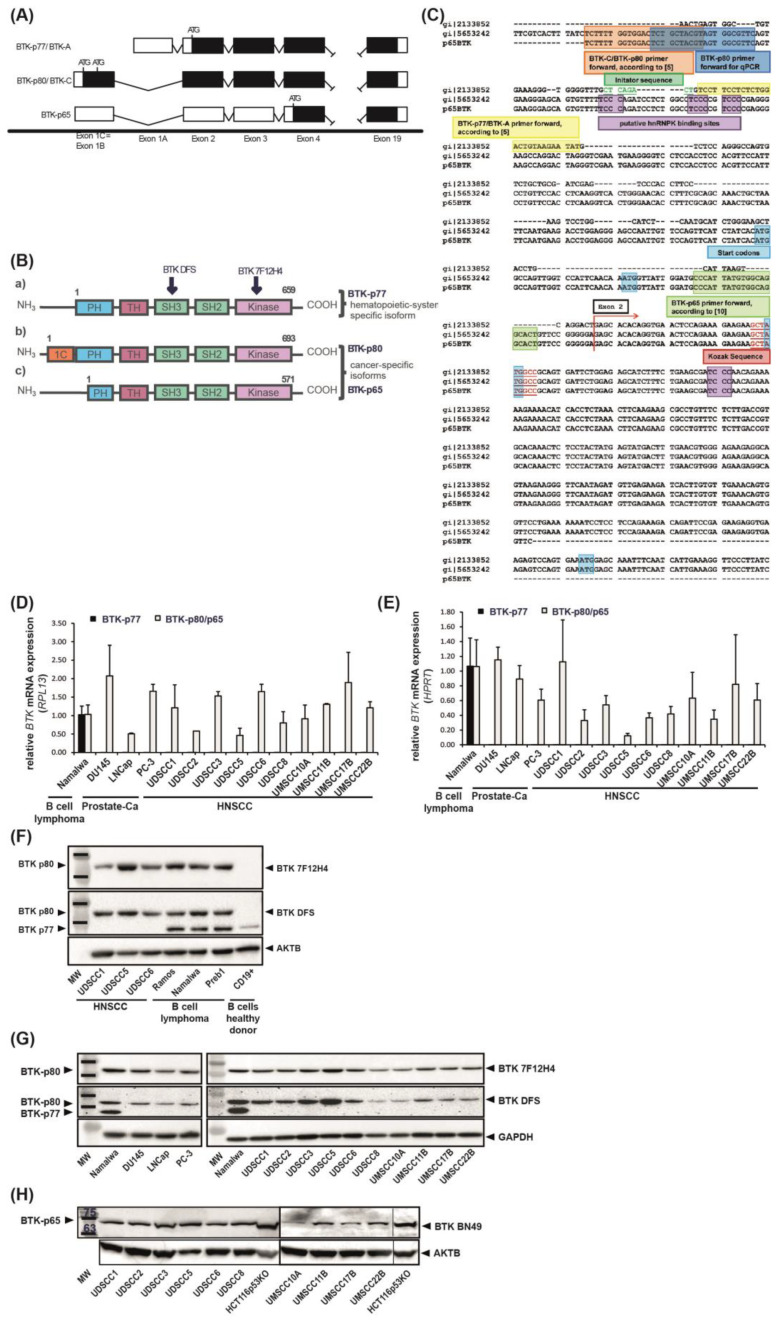
Oncogenic BTK-p80 and BTK-p65 isoforms are expressed in HNSCC. (**A**) Schematic representation showing alternative splicing of the three different BTK isoforms inferred from sequence analysis and its consequences on protein level. The BTK-p80 isoform mRNA contains an alternative first exon (exon 1C; described in Eifert et al. [8], located upstream of exon 1 from the BTK-p77 transcript) and a putative transcription start site has been described upstream of this exon 1C [8]. The first exons of BTK-p77 and BTK-p80, exon 1A and exon 1C, respectively, utilize different donor sites to splice into the same acceptor site which is located in exon 2. Exon 1C contains two start codons that are in frame with the start codon from the BTK-p77 transcript. Utilizing the first start codon leads to an extension of 34 amino acids at the N-terminus of the protein. Our analysis revealed, that BTK-p65 is expressed from the same alternative exon 1C (previously described as exon 1B by Grassilli et al. [13]). Translation of BTK-p65, however, is proposed to start from a start codon located in exon 4. (**B**) Consequently, BTK-p65 lacks most of the amino-terminal PH-domain. The three mRNA sequences are identical downstream from exon 2. Arrows indicate binding sites of BTK DFS and BTK 7F12H4 antibodies, respectively. Black/white boxes indicate translated/untranslated exons. ATG, start codon; PH, pleckstrin homology domain; TH, Tec homology domain; SH: Src homology interaction domain. (**C**) Nucleotide sequence of the published BTK-p77 isoform (accession: NM_000061.2, here gi|2133852) was aligned to the sequence of BTK-p80 (accession: NM_001287344, here gi|5653242) [8] and BTK-p65 [13] using ClustalX 2.1. Sequence alignment revealed that the BTK-p80 and the BTK-p65 sequence are identical. Therefore, the described exons 1B [13] and 1C [8] are the same. All three isoforms are identical downstream from the beginning of exon 2. Primers used for qRT-PCR by Eifert et al. [8] to detect BTK-p80 (orange) and by Grassilli et al. [13] (green) to detect BTK-p65 as well as the primers applied in this paper (blue) are highlighted. Important gene sequences associated with transcription and translation (i.e., initiator sequence, start codon, hnRNPK binding sites and Kozak sequence) are highlighted, too. hnRNPK, homo sapiens heterogeneous nuclear ribonucleoprotein K. (**D**,**E**) Quantitative RT-PCR analyses of BTK-p77 and BTK-p80/p65 mRNA expression in ten HNSCC-derived cell lines relative to two different housekeeping genes (RPL13, (**D**) and HPRT, (**E**)). Three prostate-carcinoma cell lines (DU145, LNCap and PC-3) and the Burkitt lymphoma cell line Namalwa were included into analyses. BTK mRNA expression levels of tumor cell lines are depicted relative to the BTK transcript level of Namalwa cells. Experiments were performed three times in duplicates. Presented are the mean values ± SD. (**F**) Evaluation of isoform-specificity of commercially available BTK antibodies by immunoblot analysis. BTK-p80 was detected by using two different antibodies with different binding sites (BTK 7F12H4 binding to amino acids 459–659 of BTK kinase and BTK DFS binding to SH3 domain). BTK DFS additionally detected the hematopoietic BTK-p77 isoform in healthy B cells. The detection of AKTB (β-actin) served as loading control. (**G**) Evaluation of BTK-p77 and BTK-p80 expression by immunoblot analysis. BTK-p80 was detected by using the BTK 7F12H4 and the BTK DFS antibodies. Detection of GAPDH served as loading control. (**H**) Evaluation of BTK-p65 expression by immunoblot analysis. BTK-p65 was detected by using the recently published BTK antibody BN49 [12]. The detection of AKTB (β-actin) served as loading control.

**Figure 2 cancers-15-00310-f002:**
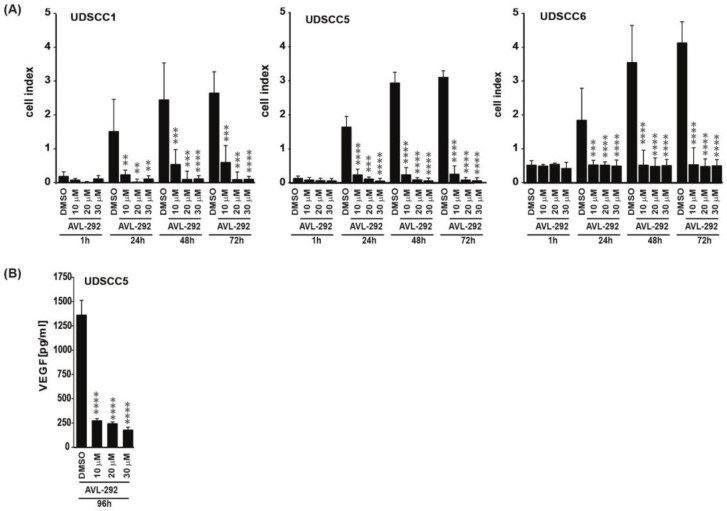
BTK inhibition affects transmigration and VEGFA secretion in HNSCC-derived cell lines. (**A**) The ability of BTK inhibition to inhibit transmigration was evaluated using the xCELLigence real time cell analysis system. The electrodes of transwells were coated with fibronectin. 20.000 cells/well were seeded. The BTK inhibitor AVL-292 was added in different concentrations as indicated. DMSO treatment served as internal control. Transmigration was monitored over time. Three independent experiments for each cell line were performed in quadruplicates. Depicted are the mean values ± SD. (**B**) Inhibition of BTK activity reduces VEGFA secretion of treated HNSCC-derived cancer cells. UDSCC5 cells were treated with different concentrations of AVL-292 for 96 h. VEGFA concentration was determined by ELISA. Three independent experiments were performed in duplicates. Depicted are the mean values ± SD ** *p* < 0.01, *** *p* < 0.001, **** *p* < 0.0001.

**Figure 3 cancers-15-00310-f003:**
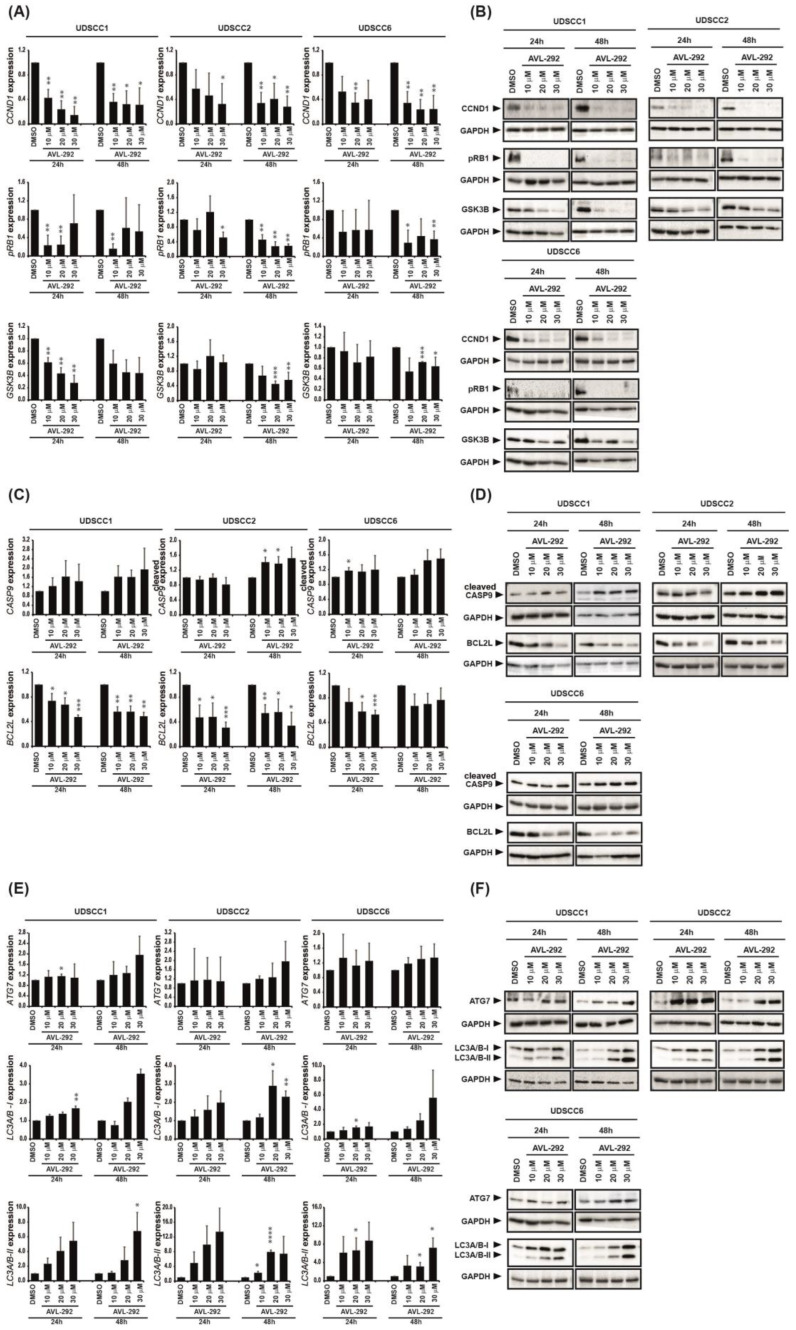
BTK inhibition induces cell cycle arrest, apoptosis and autophagy in HNSCC-derived cell lines. (**A–F**) UDSCC1, UDSCC2 and UDSCC6 were either left untreated (DMSO control) or treated with different concentrations (10, 20 or 30 µM) of AVL-292. After 24 h or 48 h, lysates were prepared and subjected to immunoblot analysis detecting proteins involved in cell cycle (**A**,**B**), apoptosis (**C**,**D**) or autophagy (**E**,**F**) as indicated. Presented are mean values ± SD of respective protein expression relative to the housekeeping protein GAPDH, determined by densitometry analyses of three independent experiments (**A**,**C**,**E**). Representatives immunoblot images of analyzed proteins are shown in B, D and F. * *p* < 0.05, ** *p* < 0.01, *** *p* < 0.001, **** *p* < 0.0001.

**Figure 4 cancers-15-00310-f004:**
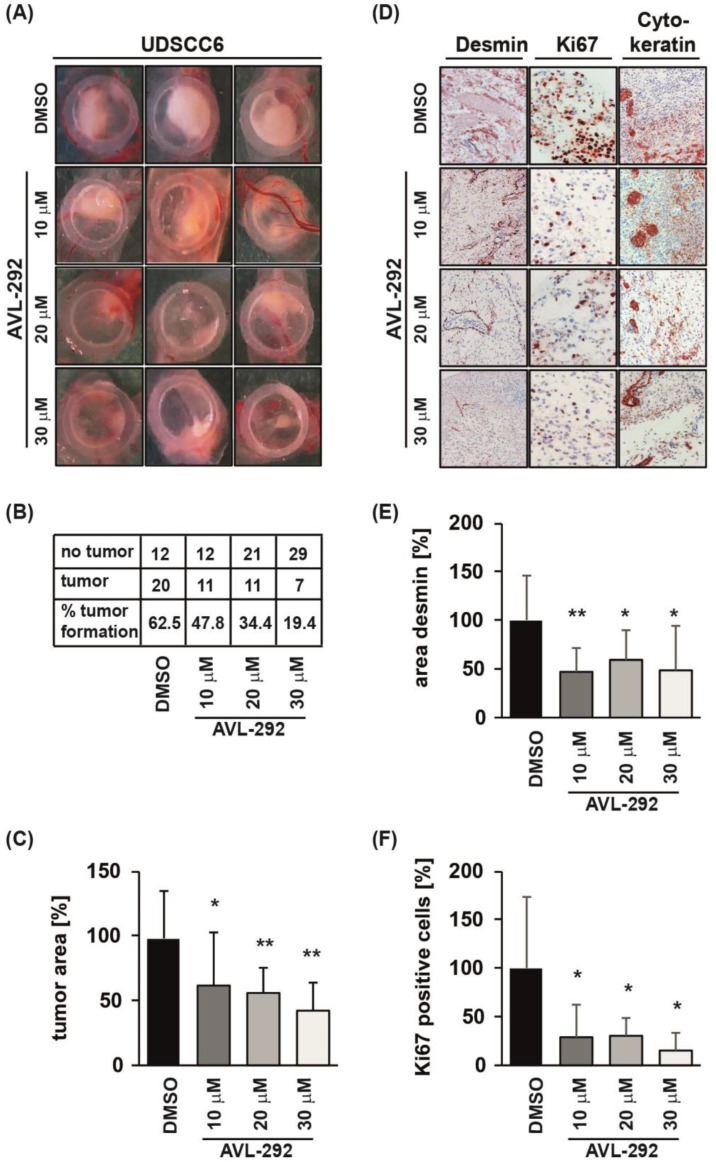
BTK inhibition impairs tumor growth and angiogenesis in vivo. (**A**) UDSCC6 cells were applied to chicken CAM. After 24 h and 48 h tumors were either treated with DMSO as negative control or with different concentrations (10, 20 or 30 µM) of AVL-292. Four days after implantation tumors were harvested, photographed, paraffin embedded and analyzed. Inhibition of BTK activity reduces (**B**) tumor formation and (**C**) tumor size. (**D**) Tumors were stained for desmin, Ki67 and cytokeratin immunohistochemically. Representative images are shown. (**E**) To quantify desmin-positive areas, background staining was subtracted from specific desmin signal using ImageJ software. (**F**) Quantification of cells stained positive for proliferation marker Ki-67 is shown. To determine the number of Ki-67-positive cells, at least 140 (up to 1270) cells in three distinct, randomly chosen areas within the tumor (according to the cytokeratin staining) were examined. In (**C**,**E**,**F**) the mean values ± SD are presented. Significancywas determined relative to DMSO control values. * *p* < 0.05, ** *p* < 0.01.

**Figure 5 cancers-15-00310-f005:**
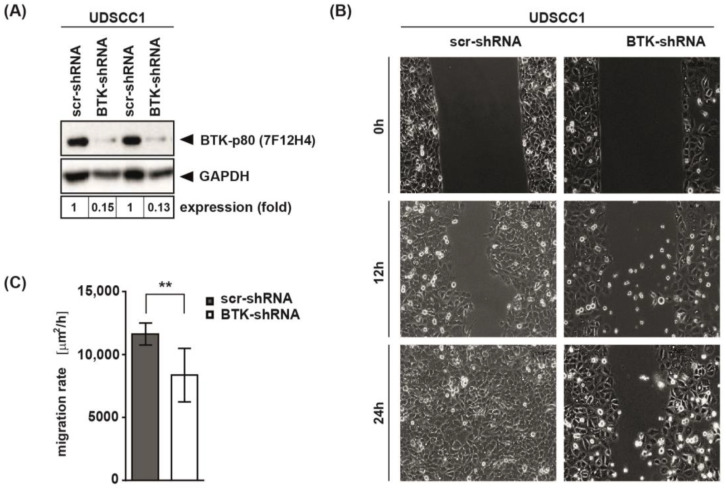
Genetic abrogation of BTK expression delayed wound closure of HNSCC-derived cell lines. (**A**) BTK was knocked-down in UDSCC1 cells. UDSCC1 cells were transduced with lentiviruses with vectors expressing either scrambled shRNA (scr-shRNA) or shRNA targeting BTK (BTK-shRNA). The efficiency of BTK knock-down was analyzed by immunoblot using antibody against BTK-p80 (7F12H4). Detection of GAPDH served as loading control. (**B**,**C**) Wound closure capacities were analyzed for scr-shRNA or BTK-shRNA transduced UDSCC1 cells. Wound closure was monitored microscopically for 24 h. Representative images for wound closure assay are depicted. (**C**) The migration rate of scr-shRNA and BTK-shRNA transduced UDSCC1 cells was calculated with the differences of the wound area measured over time. Presented are the mean values ± SD. ** *p* < 0.01.

**Figure 6 cancers-15-00310-f006:**
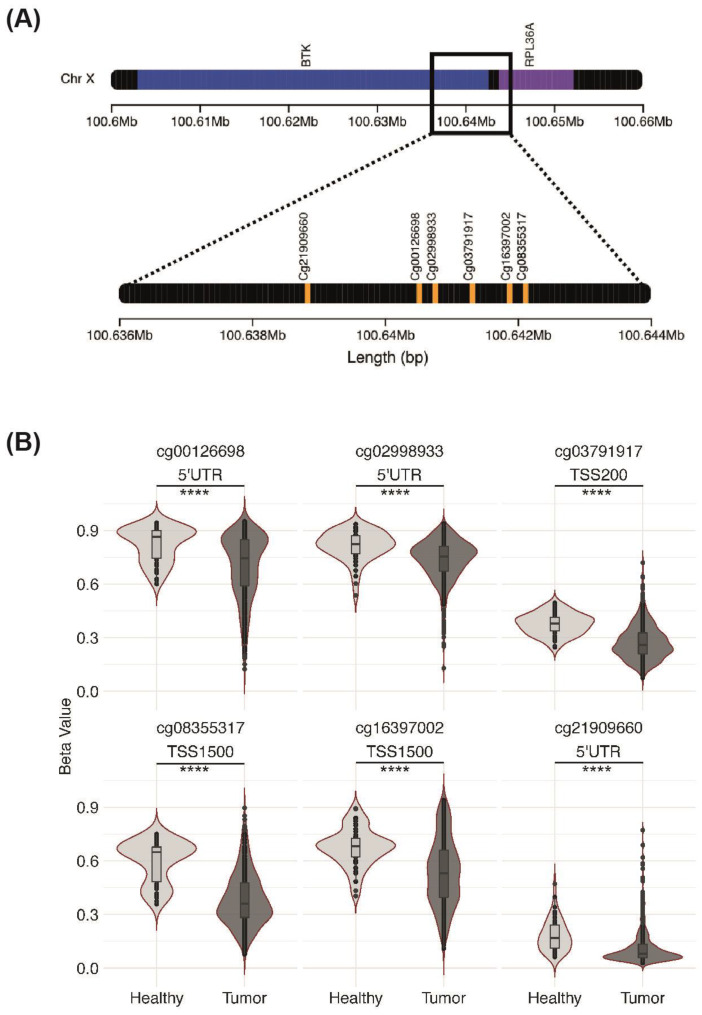
BTK-p80/p65 alternative promoter CpG loci methylation. (**A**) Chromosomal mapping of BTK (blue) and RPL36 (violet) genes located on the X chromosome. CpG loci mapped to the BTK alternative promoter are depicted in orange. (**B**) Violin plots displaying differences in methylation levels (Beta Values) of six CpG loci between TCGA healthy mucosa (n = 50) and HNSCC tissue (n = 528); **** FDR < 0.0001.

## Data Availability

The data presented in this study are available on request from the corresponding author.

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
