# Peer review of "BTK Isoforms p80 and p65 Are Expressed in Head and Neck Squamous Cell Carcinoma (HNSCC) and Involved in Tumor Progression"

_cancers, 2023, doi:10.3390/cancers15010310_

Round 1

Reviewer 1 Report

The authors aimed to investigate the expression of BTK isoforms in HNSCC and also the consequences of BTK expression for tumorigenesis. They successfully described the expression of BTK-p80/65 isoforms in several SCC cell lines as well as in primary HNSCC samples. Their data suggests that methylation might be responsible for the initiation of BTK-p80/p65 expression in HNSCC. As with other solid tumors, inhibition of BTK activity in HNSCCs models hampered proliferation and vascularization, and was associated with cell cycle arrest and increased apoptosis and autophagy. 

The study is well structured and combines patient data with in vitro and in vivo experiments and provides new knowledge on HNSCCs that can used envision novel beneficial therapeutic options for these kinds of cancers.

Minor remarks:

In the abstract section (lines 28 and 36) the authors claim to have revealed that both BTK-p80/65 isoforms are products of the same mRNA. I believe this information was previously reported for colon cancer cells in an earlier work from the same group and was confirmed here for HNSCC.

Line 226 – Healthy B cells isolated from human PBMCs were not mentioned in the material and methods (M&M) section. 

In Figure 1F – Ramos and PreB cells were not mentioned in M&M section.

Did the authors investigate p65 expression on Namalwa cells?

In Figure S1 the antibody used was mentioned only in the original image file. Also, human samples (also mentioned in lines 235 and 236) were not described in the M&M section.

Although described in the figure legend ±SDs were not represented in the figure S2A graphs.

Cell cycle analysis (Fig S2 B-E) was not described in M&M section.

There is a configuration problem in figure 3D, UDSCC2 48h.

In line 375 and in figure 5 legend the authors state that BTK inhibition abolishes migration. Although, it delayed wound closure in the scratch assay, it clearly did not abrogate migration. Also, how did the authors correct for proliferation?   

Author Response

Reviewer 1

Minor remarks:

In the abstract section (lines 28 and 36) the authors claim to have revealed that both BTK-p80/65 isoforms are products of the same mRNA. I believe this information was previously reported for colon cancer cells in an earlier work from the same group and was confirmed here for HNSCC.

Thank you very much for taking time to review our manuscript. It has not been shown in advance in an original article that both BTK-p80 and p65 isoforms are products of the same mRNA. Alignment of BTK-p80 and p65 sequences was done in the here presented work for the first time. In 2013, the p80 mRNA was first described by Eifert et al. in breast cancer. In 2016, the p65 mRNA was first described by Grassilli et al. in colon cancer. It was only suggested in a review article that both mRNAs might be identical but both studies were conducted independently. However, confirmation that both mRNAs are identical is shown in our manuscript for the first time. Additionally, here we report for the first time, that bot oncogenic Btk isoforms, BTK-p80 and BTK-p65 are co-expressed in cancer cells.

Line 226 – Healthy B cells isolated from human PBMCs were not mentioned in the material and methods (M&M) section.

Missing information about isolated healthy B cells from human PBMC is now added in the M&M section and in table S3.

In Figure 1F – Ramos and PreB cells were not mentioned in M&M section.

The required information was added under “Cell lines and cell culture” in M&M section.

Did the authors investigate p65 expression on Namalwa cells?

No, we did not investigate the expression of BTK-p65 on Namalwa cells.

In Figure S1 the antibody used was mentioned only in the original image file. Also, human samples (also mentioned in lines 235 and 236) were not described in the M&M section.

We agree that this information is definitely missing. We indicated which antibody was used both in the figure itself and also in the figure legend. Information about human samples is now added to the M&M section.

Although described in the figure legend ±SDs were not represented in the figure S2A graphs.

This mistake was now corrected in the figure legend.

Cell cycle analysis (Fig S2 B-E) was not described in M&M section.

Required information was now added to the M&M section.

There is a configuration problem in figure 3D, UDSCC2 48h.

Thank you very much for this hint. The problem in Figure 3D is corrected now.

In line 375 and in figure 5 legend the authors state that BTK inhibition abolishes migration. Although, it delayed wound closure in the scratch assay, it clearly did not abrogate migration. Also, how did the authors correct for proliferation?  

We agree with this comment and changed the corresponding positions in the text. We agree that we should rather refer to wound closure or proliferation instead of migration and changed this accordingly. Nevertheless, we could show that BTK inhibition results in significantly reduced transmigratory capacity (Transwell assay, Figure 2A).

Reviewer 2 Report

This study aims to explore the relationship of Bruton’s Tyrosine Kinase (BTK) isoforms of 80 and 65 kDa in in head and neck squamous cell carcinoma (HNSCC) and further the molecular and cellular consequences of BTK expression for HNSCC tumorigenesis. This study shows that both isoforms are products of the same mRNA. The epigenetic process of methylation might be responsible for the initiation of BTK-p80/p65 expression in HNSCC. In addition, the chemical or genetic abrogation of BTK activity leads to inhibition of tumor progression in terms of proliferation and vascularization in vitro and in vivo and these were associated with cell cycle arrest and increased apoptosis and autophagy. These data indicate BTK-p80 and BTK-p65 as novel HNSCC-associated oncogenes. Abundant BTK expression is a characteristic feature of primary and metastatic HNSCC, targeting BTK activity appears as a promising therapeutic option for HNSCC patients. This manuscript content is suitable for publication and related studies still not publish. However, there are some research results need further explain in this manuscript.

1.        Some errors in the manuscript content need to be corrected.

Author Response

Reviewer 2

Comments and Suggestions for Authors

This study aims to explore the relationship of Bruton’s Tyrosine Kinase (BTK) isoforms of 80 and 65 kDa in in head and neck squamous cell carcinoma (HNSCC) and further the molecular and cellular consequences of BTK expression for HNSCC tumorigenesis. This study shows that both isoforms are products of the same mRNA. The epigenetic process of methylation might be responsible for the initiation of BTK-p80/p65 expression in HNSCC. In addition, the chemical or genetic abrogation of BTK activity leads to inhibition of tumor progression in terms of proliferation and vascularization in vitro and in vivo and these were associated with cell cycle arrest and increased apoptosis and autophagy. These data indicate BTK-p80 and BTK-p65 as novel HNSCC-associated oncogenes. Abundant BTK expression is a characteristic feature of primary and metastatic HNSCC, targeting BTK activity appears as a promising therapeutic option for HNSCC patients. This manuscript content is suitable for publication and related studies still not publish. However, there are some research results need further explain in this manuscript.

  1. Some errors in the manuscript content need to be corrected.

Thank you for taking time to review our manuscript. The content of the manuscript regarding introduction, M&M as well as results was improved. Introduction now includes more detailed information about BTK and HNSCC. Missing details regarding M&M including detailed information about cell lines and patient samples as well as CD19+ B cell isolation and cell cycle analysis are now included. In the results section it is now further explained why we chose certain cell lines for certain experiments.

Reviewer 3 Report

In this study entitled "BTK isoforms, p80 and p65 are expressed in head and neck squamous cell carcinoma (HNSCC) and involved in tumor progression" Authors tried to explain the importance of oncogenic  BTK-p65  and  BTK-p80 isoforms in HNSCC.

I have some major comments,

1. Introduction is not well written. The authors failed to properly introduce BTK. No mention of HNSCC,(HPV+ OR NEGATIVE). If you are focusing on HNSCC, You should mention both groups.

2. What were the criteria for selecting the cell lines for the study (randomly selected?)

Provide the details of the cell lines and patient samples used (HPV status, mutation status etc. As a table)

3. Figure 1H.The WB data is not convincing(even the supplementary data) How can you cut and paste from different blots? Should provide a new data

4. Authors must provide invivo (mice xenograft) data for at least 2 cell lines. Chicken CAM experiment in one cell line is not sufficient. Why did you choose only UDSCC6?.

Author Response

Reviewer 3

I have some major comments,

  1. Introduction is not well written. The authors failed to properly introduce BTK. No mention of HNSCC,(HPV+ OR NEGATIVE). If you are focusing on HNSCC, You should mention both groups.

Thank you very much for taking time to review our manuscript. The introduction is now changed accordingly. More detailed information regarding BTK and HNSCC is provided now.

  1. What were the criteria for selecting the cell lines for the study (randomly selected?)

The cell lines were not randomly selected. Cell lines were selected based on their BTK-p80 expression and their responsiveness to BTK inhibitors revealed by the proliferation assay. We aimed to include cell lines that cover the whole spectrum of Btk expression and responsiveness to BTK inhibitors. Consequently, we used UDSCC1 and UDSCC2 (intermediate BTK expression, intermediate responder), UDSCC5 (high BTK expression, strong responder) and UDSCC6 (lower BTK expression, low responder) for subsequent experiments. This information is now also included in the Results section.

Provide the details of the cell lines and patient samples used (HPV status, mutation status etc. As a table)

These details are now provided in Table S1-S3.

  1. Figure 1H.The WB data is not convincing(even the supplementary data) How can you cut and paste from different blots? Should provide a new data

WBs from Figure 1H and Figure S1 were not cut and paste from different blots. All samples shown in these figures ran on the same gel/plot. We also uploaded the original WB files where this is evident. It is true that 2 samples were excluded from the blot shown in Figure 1H, as we did not own the removed cell lines and were not allowed to show this data. Also, one sample was excluded from the blot shown in Figure S1, as tumor tissue #3 also did not reveal any beta-actin expression indicating that no protein could be isolated from this tissue. But this is all traceable by means of the provided original WB images.

  1. Authors must provide in vivo (mice xenograft) data for at least 2 cell lines. Chicken CAM experiment in one cell line is not sufficient. Why did you choose only UDSCC6?.

CAM assay is a well-established in vivo model and is an accepted alternative model to refine animal experiments in terms of reducing pain or suffering of experimental animals. It is a well-established model for the investigation of angiogenesis and tumor growth in response to chemical or genetical abrogation of kinase activity or to drugs in general, especially in the field of cancer research. One great advantage of CAM for our experimental approach was that we could transplant and grow (three-dimensional) human HNSCC tumor cells. By using CAM assay as an animal refinement model we also follow the guidelines of the journal ‘Cancers’ adhering to the commonly-accepted 3Rs. UDSCC6 was chosen because these cells revealed rather low BTK expression and low responsiveness to BTK inhibition in proliferation assay. Consequently, we chose this cell line to demonstrate that even ‘low responding’ cells to BTK inhibition reveal impaired tumor growth and angiogenesis in CAM upon treatment. This information is now also provided in the main text.

Round 2

Reviewer 3 Report

The Authors have addressed most of my concerns with the original manuscript. The revised manuscript can be accepted for publication. Good luck